# Assessment of the willingness of dentists in the state of Indiana to administer vaccines

**Anubhuti Shukla**[1]*, **Kelly Welch**[2], **Alessandro Villa**[3]

**1** Department of Cariology, Operative Dentistry and Dental Public Health, Indiana University School of Dentistry, Indianapolis, Indiana, United States of America, **2** Team Maureen, North Falmouth, Massachusetts, United States of America, **3** Department of Orofacial Sciences, University of California San Francisco School of Dentistry, San Francisco, California, United States of America

* anshukla@iu.edu

## Abstract

### Background

Human Papillomavirus associated oropharyngeal cancers have been on the rise in the past three decades. Dentists are uniquely positioned to discuss vaccination programs with their patients. The goal of this project was to assess the readiness of dentists in the state of Indiana in being able to administer vaccines.

### Methods

An 18-question online survey was sent to licensed dentists in the state of Indiana. Mantel-Haenszel chi-square tests, followed by multivariable analyses using ordinal logistic regression were conducted to assess providers' comfort levels and willingness to administer vaccines in both children and adults, by provider characteristics (practice type, location, and years in practice).

### Results

A total of 569 completed surveys were included for data analyses. Most dentists (58%) responded positively when asked if they would consider offering vaccinations in their office, if allowed by state legislation. In general, dentists working in academic settings and federally qualified health centers were more agreeable to offering vaccination in their practice. The level of agreement with "Dentists should be allowed to administer HPV, Influenza, Hep A and COVID 19 vaccines" for both children and adults decreased with increased years of practice. More than half of the respondents (55%) agreed that dental providers were competent to administer vaccines and needed no further training.

### Conclusion

The study results suggest the willingness of dentists in the state of Indiana to offer vaccinations in their practices, if allowed by legislation.

**Data Availability Statement:** All relevant data are within the manuscript and its Supporting Information files.

**Funding:** The project was supported by the Delta Dental Foundation. https://www.deltadentalin.com/

giving-back The grant was awarded to Principal Investigator for the project, Dr Anubhuti Shukla (A. S.). The funders had no role in study design, data collection and analysis, decision to publish, or preparation of the manuscript.

**Competing interests:** The authors have declared that no competing interests exist.

## Practical implications

Dental providers can be a unique resource to add to workforce for improving vaccination efforts.

## Background

The importance of vaccines can be realized in this continuing COVID -19 pandemic more than ever. To ramp up the vaccination rates, the seventh amendment to the Public Readiness and Emergency Preparedness (PREP) Act declaration has included dentists and qualified dental students in the pool of vaccinators [1]. Although this is an emergency authorization specific to the COVID-19 vaccine during the current contagion, this may be an opportune time to consider vaccination as a more permanent addition to the scope of practice for dental professionals. According to a recent survey conducted by the World Dental Federation (FDI), in one-third of the countries that responded, dentists were granted authorization to administer COVID-19 vaccines; several of those countries were the ones where dentists had not been previously allowed to administer any vaccine [2].

Research shows that with adequate training, dentists are willing and able to offer support for vaccine advocacy [3–5]. In 2017, more than 31.1 million people in the U.S. sought care from a dentist, but not from their physician [6]. Dentists therefore could leverage this exclusive advantage and discuss and advocate for vaccination. Of note, several states already allow dentists to administer a variety of vaccines to the public. In 2019, Oregon passed a bill that allows licensed dental providers to prescribe and administer any vaccine [7]. In Minnesota and Illinois, dentists are authorized to administer the influenza vaccine after adequate training [8]. In addition, during the H1N1 epidemic in recent years, dentists were authorized to administer vaccines against H1N1 to assist in the frontline pandemic response [9]. Even prior to the PREP Act amendment, more than twenty-one states had already issued emergency authorization to allow licensed dental providers to administer the COVID-19 vaccine as part of their vaccination roll out plans [10]. While it would be ideal for dentists to have the authorization to administer any vaccine, our current study focuses only on influenza, COVID-19, hepatitis A, and human papillomavirus (HPV) vaccines.

While the entire world's leaders are engrossed in making the COVID-19 vaccination available to all, our study focuses on some additional vaccines beyond COVID-19 for particular reasons. The influenza vaccine is included because there are states that already allow dentists to administer it, and there is historical precedence (H1N1) for broad use of dental providers as influenza vaccinators [11]. Hepatitis vaccination was included because of the current hepatitis issues in Indiana, where the study was conducted. Since November 2017, the Indiana State Department of Health (ISDH) has been investigating an outbreak of acute hepatitis A virus (HAV). There have been almost 2471 cases with 1376 hospitalizations as of November 2020, demonstrating a need for expanding vaccination opportunities for hepatitis [12].

Recent data from the Centers for Disease Control and Prevention (CDC) reports an annual average incidence of nearly 45,300 HPV-associated cancer cases, including about 25,400 in women, and about 19,900 in men [13]. Cervical cancer is the most common HPV-associated cancer among women, and oropharyngeal cancer is the most common in men [13]. Oropharyngeal cancers in Western Countries have been on the rise in the past three decades, mainly due to an increase in persistent high-risk-type human papillomavirus (HPV) infections [14–17]. Most HPV-related cancers are preventable through HPV vaccination (Gardasil 9® HPV

9 valent-vaccine, recombinant; Merck & Co., Inc.) [18, 19], yet the rates of HPV vaccination remain low as compared to other adolescent vaccinations. Pre-COVID 19 pandemic averages of HPV vaccine completion rates in the US were low (54.2%) [20] and with the pandemic, the rates of all immunizations, including HPV vaccinations, have dropped even more, putting millions of children at risk for developing persistent HPV infections, genital warts, and life-threatening HPV-related cancers later in life [21]. This decrease in vaccination rates further drives the need to have more channels to aid in vaccine distribution and build vaccine confidence. Since dental providers screen their adult patients annually for oral cancers, and given the established association between HPV and oropharyngeal cancers, these providers should be able to administer HPV vaccinations to their younger patients to provide long-term HPV protection [22, 23].

The concept of non- traditional vaccinators has been long discussed; however, the current COVID-19 pandemic has further highlighted the importance of expanding the network of vaccine providers. In the recent past, including pharmacists as vaccine administrators has shown to improve vaccination coverage rates, reduce costs, and has been well-received both by the patient population and pharmacists alike [24–28]. Expanding dental providers scope of practice to include vaccination could provide similar results as with pharmacists, however such changes nationwide would call for legislative enabling, educational trainings and professional indemnity. The goal of this study is to explore the willingness of dental providers in the state of Indiana to offer vaccination in their practice, if allowed by legislation.

## Methods

### Study population and survey instrument

The intent was to send out a web-based survey developed on the Qualtrics platform to all the dentists with an active license in the state of Indiana. To accomplish this goal, different organization listservs were used to reach our sample population. The survey e-mails were sent out in third week of October 2020, with reminders being sent out every month until February 28[th] 2021, after which the survey was closed. All survey participants received a gift card ($5 value) to Amazon. The study was supported by Delta Dental Foundation of Indiana. The Institutional Review Board at the hosting institution approved this study (exempt; IRB Study# 2010179094) as no identifiable data was collected from any participants. This designation waived the IRB need for formal written consent. Respondents read a statement which informed them that the study was completely voluntary, summarized the purpose of the study, and how their data would be used. Respondents were informed that by completing the survey they were offering their consent to participate.

The 18-question survey was internally validated amongst the research investigators on the project, along with feedback from subject matter experts. The main variable of interest was: dentists' willingness to offer vaccination in their practice, if allowed by legislation. Data on location of practice (rural/urban), type of practice setting, and number of years in practice were collected with multiple-choice questions. A 5-point Likert-type scale ranging from "Strongly Agree" to "Strongly Disagree" was used for participants' agreement to statements around their perception toward vaccination in general, dentists administering vaccines, and their comfort level in administering vaccines to children and adults. Questions assessing major challenges in dentists agreeing to offer vaccinations in their practice were also included as multiple choice. Additional comments were recorded as responses for open ended questions.

The details of all questions on the survey can be found under S1 File.

## Data analyses

Missing data from incomplete survey responses were excluded from the analysis.

Descriptive statistics were used to evaluate the distribution of participant characteristics by type of practice setting, location of practice, and years in practice. Frequency analysis for the responses to the Likert scale questions were also calculated. Spearman correlations were calculated to evaluate the associations among the responses to the Likert scale questions.

Chi-square tests were used to evaluate the individual associations of each participant characteristic with whether the provider would consider offering vaccinations, followed by multivariable analyses using logistic regression. Mantel-Haenszel chi-square tests for ordered categorical data were used to evaluate the individual associations of each participant characteristic with the Likert-scale agreement ratings for the beliefs, attitudes, and comfort levels around vaccination items, followed by multivariable analyses using ordinal logistic regression with a cumulative logit link. In the multivariable analyes, the ordinal years in practice variable was treated as a continuous variable. When the overall tests for type of practice were significant, pair-wise tests among the practice types were conducted. A generalized linear mixed model for cumulative logistic regression with a random effect to account for within-subject correlation was used to compare responses between questions. A 5% significance level was used for all tests. The data was analyzed using SAS version 9.4 [SAS Institute, Inc., Cary, NC, USA].

## Results

A total of 622 dentists responded to the survey. After excluding dentists with incomplete responses, a total of 569 completed surveys were included for data analyses.

### Practice type, location of practice, and years in practice [Table 1]

[Table 1] shows descriptive statistics of the responses for type of practice, years in practice, and location of practice. More than half the respondents (66%) were from private practice

**Table 1. Participant characteristics.**

| Characteristics | N (%) |
|---|---|
| **Practice setting** | |
| Private Practice | 373 (66%) |
| Academic Institution | 83 (15%) |
| FQHC and similar * | 73 (13%) |
| DSO ** | 18 (3%) |
| Other *** | 22 (4%) |
| **Years of clinical practice** | |
| 0–5 years | 99 (17%) |
| 6–10 years | 70 (12%) |
| 11–15 years | 70 (12%) |
| 16–20 years | 58 (10%) |
| > 21 years | 272 (48%) |
| **Location of clinical practice** | |
| Rural | 154 (27%) |
| Urban | 415 (73%) |

FQHC* = Federally qualified health center

DSO** = Dental Service Organization

Others*** = Hospital Based Clinic, Local Health Department, Mobile Dentistry Practice, Other (not specified)

**Table 2. Beliefs, attitudes, and comfort levels around vaccination.**

| | Strongly Agree | Agree | Neither | Disagree | Strongly Disagree |
|---|---|---|---|---|---|
| | N (%) | N (%) | N (%) | N (%) | N (%) |
| There is scientific proof that immunization prevents infectious diseases | 441 (78%) | 97 (17%) | 7 (1%) | 5 (1%) | 19 (3%) |
| Everyone should be receiving the recommended vaccinations (excluding those with prohibiting medical conditions) | 291 (51%) | 178 (31%) | 38 (7%) | 34 (6%) | 28 (5%) |
| Given the COVID-19 pandemic, if the Indiana State Board of Dentistry authorizes dentists to provide vaccinations, would you consider offering vaccination in your practice? | 176 (31%) | 174 (31%) | 101 (18%) | 46 (8%) | 72 (13%) |
| Dental providers are competent enough to be able to administer vaccines and need no further education/training | 155 (27%) | 158 (28%) | 94 (17%) | 125 (22%) | 37 (7%) |
| I am comfortable administering vaccines in children | 58 (10%) | 114 (20%) | 136 (24%) | 139 (24%) | 122 (21%) |
| I am comfortable administering vaccines in adults | 138 (24%) | 167 (29%) | 151 (27%) | 64 (11%) | 49 (9%) |
| Dentists should be allowed to administer HPV, Influenza, Hep A and COVID 19 in children | 96 (17%) | 146 (26%) | 192 (34%) | 77 (14%) | 58 (10%) |
| Dentists should be allowed to administer vaccines such as HPV, Influenza, Hep A and COVID 19 in adults. | 158 (28%) | 188 (33%) | 135 (24%) | 47 (8%) | 41 (7%) |
| HPV related Oro-pharyngeal cancers can be prevented by use of vaccines | 240 (42%) | 214 (38%) | 98 (17%) | 6 (1%) | 11 (2%) |
| It would be easier for patients to complete their HPV vaccine schedule if they were to receive it from their dentists | 127 (22%) | 195 (34%) | 177 (31%) | 36 (6%) | 34 (6%) |

settings. Of all the respondents, almost half (48%) reported being in practice for 21 years or longer. The majority of the participants (73%) had their practice located in an urban (non-rural) location. More than half of the respondents (68%) reported having a policy for oral cancer screening in their office (data not reported).

## Beliefs and perceptions [Table 2]

The distribution of responses to the questions that centered around beliefs, perception, and comfort levels of the participants towards vaccination in general, and in administering specific vaccines such as HPV, influenza, hepatitis A and COVID-19 is reported in [Table 2]. More than half of the respondents (55%) agreed that dental providers were competent to administer vaccines, and needed no further education/training. A majority of respondents (62%) reported that they would consider offering the COVID-19 vaccination in their practice, if the Indiana State Board of Dentistry authorized dentists to provide vaccinations (emergency order, etc.) during the pandemic.

## Consider the idea of offering vaccination in their practice [Table 3]

The majority of dentists (58%; N = 331) responded positively when asked if they would consider offering vaccinations in their office, if allowed by state legislation. There were no significant associations of provider characteristics like practice setting, area of practice and years of practice with dentists' willingness to vaccinate in their practice.

## Practice characteristic associations with beliefs and perceptions [Table 4]

The level of agreement with "I am comfortable administering vaccines in children" decreased with increased years of practice. Dentists reported being more comfortable vaccinating adults than children. Dentists working at Dental Service Organizations (DSO)s had a lower level of

**Table 3. Association of participant characteristics and vaccinations.**

| Question: Would you consider offering vaccinations in your practice, if allowed by legislation? | | | | p-values | |
| --- | --- | --- | --- | --- | --- |
| | | Yes | No | Single-variable | Multivariable |
| TOTAL | | 331 (58%) | 238 (42%) | | |
| Characteristics | | | | | |
| Practice Setting | Private Practice | 214 (57%) | 159 (43%) | 0.674 | 0.650 |
| | Academic Institution | 54 (65%) | 29 (35%) | | |
| | FQHC and similar * | 41 (56%) | 32 (44%) | | |
| | DSO ** | 9 (50%) | 9 (50%) | | |
| | Other *** | 13 (59%) | 9 (41%) | | |
| Years in Practice | 0–5 years | 59 (60%) | 40 (40%) | 0.817 | 0.779 |
| | 6–10 years | 38 (54%) | 32 (46%) | | |
| | 11–15 years | 44 (63%) | 26 (37%) | | |
| | 16–20 years | 34 (59%) | 24 (41%) | | |
| | ≥ 21 years | 156 (57%) | 116 (43%) | | |
| Location of clinical practice | Rural | 93 (60%) | 61 (40%) | 0.514 | 0.472 |
| | Urban | 238 (57%) | 177 (43%) | | |

FQHC* = Federally qualified health center

DSO** = Dental Service Organization

Others*** = Hospital Based Clinic, Local Health Department, Mobile Dentistry Practice, Other (not specified)

**Table 4. Beliefs, attitudes, and comfort levels around vaccination by participant characteristics.**

| | | | | | | p-values | |
| --- | --- | --- | --- | --- | --- | --- | --- |
| Predictor | Strongly Agree | AFgree | Neither Agree nor Disagree | Disagree | Strongly Disagree | Single variable | Multivariable |
| I am comfortable administering vaccines in children | | | | | | | |
| Practice Setting | | | | | | | |
| Private Practice | 32 (9%) | 63 (17%) | 96 (26%) | 100 (27%) | 82 (22%) | 0.001 | 0.001 |
| Academic Institution | 14 (17%) | 23 (28%) | 13 (16%) | 16 (19%) | 17 (20%) | | |
| FQHC and similar * | 3 (4%) | 28 (38%) | 18 (25%) | 13 (18%) | 11 (15%) | | |
| DSO ** | 1 (6%) | 0 (0%) | 3 (17%) | 6 (33%) | 8 (44%) | | |
| Other *** | 8 (36%) | 0 (0%) | 6 (27%) | 4 (18%) | 4 (18%) | | |
| Years in Practice | | | | | | | |
| 0–5 years | 13 (13%) | 29 (29%) | 23 (23%) | 12 (12%) | 22 (22%) | < .001 | 0.001 |
| 6–10 years | 9 (13%) | 10 (14%) | 12 (17%) | 31 (44%) | 8 (11%) | | |
| 11–15 years | 9 (13%) | 19 (27%) | 22 (31%) | 14 (20%) | 6 (9%) | | |
| 16–20 years | 7 (12%) | 17 (29%) | 11 (19%) | 11 (19%) | 12 (21%) | | |
| ≥ 21 years | 20 (7%) | 39 (14%) | 68 (25%) | 71 (26%) | 74 (27%) | | |
| Location of clinical practice | | | | | | | |
| Rural area | 20 (13%) | 31 (20%) | 30 (19%) | 38 (25%) | 35 (23%) | 0.745 | 0.480 |
| Urban area | 38 (9%) | 83 (20%) | 106 (26%) | 101 (24%) | 87 (21%) | | |
| I am comfortable administering vaccines in adults | | | | | | | |
| Practice Setting | | | | | | | |
| Private Practice | 83 (22%) | 113 (30%) | 102 (27%) | 44 (12%) | 31 (8%) | 0.173 | 0.028 |
| Academic Institution | 28 (34%) | 26 (31%) | 12 (14%) | 8 (10%) | 9 (11%) | | |

*(Continued)*

**Table 4.** (Continued)

| | | | | | | | |
|---|---|---|---|---|---|---|---|
| FQHC and similar * | 9 (12%) | 24 (33%) | 26 (36%) | 7 (10%) | 7 (10%) | | |
| DSO ** | 8 (44%) | 4 (22%) | 2 (11%) | 3 (17%) | 1 (6%) | | |
| Other *** | 10 (45%) | 0 (0%) | 9 (41%) | 2 (9%) | 1 (5%) | | |
| **Years in Practice** | | | | | | | |
| 0–5 years | 28 (28%) | 32 (32%) | 19 (19%) | 9 (9%) | 11 (11%) | 0.213 | 0.059 |
| 6–10 years | 14 (20%) | 19 (27%) | 20 (29%) | 13 (19%) | 4 (6%) | | |
| 11–15 years | 24 (34%) | 24 (34%) | 11 (16%) | 11 (16%) | 0 (0%) | | |
| 16–20 years | 12 (21%) | 18 (31%) | 16 (28%) | 8 (14%) | 4 (7%) | | |
| ≥ 21 years | 60 (22%) | 74 (27%) | 85 (31%) | 23 (8%) | 30 (11%) | | |
| **Location of clinical practice** | | | | | | | |
| Rural area | 37 (24%) | 51 (33%) | 36 (23%) | 18 (12%) | 12 (8%) | 0.590 | 0.469 |
| Urban area | 101 (24%) | 116 (28%) | 115 (28%) | 46 (11%) | 37 (9%) | | |
| **Dentists should be allowed to administer HPV, Influenza, Hep A and COVID 19 in children** | | | | | | | |
| **Practice Setting** | | | | | | | |
| Private Practice | 52 (14%) | 97 (26%) | 127 (34%) | 55 (15%) | 42 (11%) | 0.013 | 0.004 |
| Academic Institution | 17 (20%) | 24 (29%) | 20 (24%) | 11 (13%) | 11 (13%) | | |
| FQHC and similar * | 12 (16%) | 18 (25%) | 31 (42%) | 8 (11%) | 4 (5%) | | |
| DSO ** | 9 (50%) | 3 (17%) | 5 (28%) | 0 (0%) | 1 (6%) | | |
| Other *** | 6 (27%) | 4 (18%) | 9 (41%) | 3 (14%) | 0 (0%) | | |
| **Years in Practice** | | | | | | | |
| 0–5 years | 19 (19%) | 33 (33%) | 31 (31%) | 13 (13%) | 3 (3%) | < .001 | < .001 |
| 6–10 years | 14 (20%) | 19 (27%) | 18 (26%) | 16 (23%) | 3 (4%) | | |
| 11–15 years | 18 (26%) | 20 (29%) | 20 (29%) | 7 (10%) | 5 (7%) | | |
| 16–20 years | 10 (17%) | 15 (26%) | 18 (31%) | 7 (12%) | 8 (14%) | | |
| ≥ 21 years | 35 (13%) | 59 (22%) | 105 (39%) | 34 (13%) | 39 (14%) | | |
| **Location of clinical practice** | | | | | | | |
| Rural area | 22 (14%) | 42 (27%) | 51 (33%) | 21 (14%) | 18 (12%) | 0.416 | 0.837 |
| Urban area | 74 (18%) | 104 (25%) | 141 (34%) | 56 (13%) | 40 (10%) | | |
| **Dentists should be allowed to administer vaccines such as HPV, Influenza, Hep A and COVID 19 in adults.** | | | | | | | |
| **Practice Setting** | | | | | | | |
| Private Practice | 87 (23%) | 133 (36%) | 95 (25%) | 29 (8%) | 29 (8%) | 0.102 | 0.026 |
| Academic Institution | 28 (34%) | 26 (31%) | 13 (16%) | 8 (10%) | 8 (10%) | | |
| FQHC and similar * | 24 (33%) | 19 (26%) | 20 (27%) | 7 (10%) | 3 (4%) | | |
| DSO ** | 9 (50%) | 7 (39%) | 1 (6%) | 0 (0%) | 1 (6%) | | |
| Other *** | 10 (45%) | 3 (14%) | 6 (27%) | 3 (14%) | 0 (0%) | | |
| **Years in Practice** | | | | | | | |
| 0–5 years | 37 (37%) | 28 (28%) | 20 (20%) | 11 (11%) | 3 (3%) | 0.026 | 0.020 |
| 6–10 years | 17 (24%) | 26 (37%) | 16 (23%) | 8 (11%) | 3 (4%) | | |
| 11–15 years | 25 (36%) | 26 (37%) | 9 (13%) | 8 (11%) | 2 (3%) | | |
| 16–20 years | 11 (19%) | 16 (28%) | 20 (34%) | 6 (10%) | 5 (9%) | | |
| ≥ 21 years | 68 (25%) | 92 (34%) | 70 (26%) | 14 (5%) | 28 (10%) | | |
| **Location of clinical practice** | | | | | | | |
| Rural area | 47 (31%) | 46 (30%) | 34 (22%) | 14 (9%) | 13 (8%) | 0.904 | 0.533 |
| Urban area | 111 (27%) | 142 (34%) | 101 (24%) | 33 (8%) | 28 (7%) | | |
| **There is scientific proof that immunization prevents infectious diseases** | | | | | | | |
| **Practice Setting** | | | | | | | |
| Private Practice | 282 (76%) | 68 (18%) | 6 (2%) | 0 (0%) | 17 (5%) | 0.522 | 0.306 |
| Academic Institution | 64 (77%) | 15 (18%) | 0 (0%) | 4 (5%) | 0 (0%) | | |

*(Continued)*

**Table 4.** (Continued)

| | | | | | | | |
|---|---|---|---|---|---|---|---|
| FQHC and similar * | 61 (84%) | 10 (14%) | 0 (0%) | 1 (1%) | 1 (1%) | | |
| DSO ** | 14 (78%) | 3 (17%) | 1 (6%) | 0 (0%) | 0 (0%) | | |
| Other *** | 20 (91%) | 1 (5%) | 0 (0%) | 0 (0%) | 1 (5%) | | |
| **Years in Practice** | | | | | | | |
| 0–5 years | 78 (79%) | 18 (18%) | 0 (0%) | 0 (0%) | 3 (3%) | 0.811 | 0.284 |
| 6–10 years | 47 (67%) | 17 (24%) | 0 (0%) | 3 (4%) | 3 (4%) | | |
| 11–15 years | 56 (80%) | 12 (17%) | 0 (0%) | 0 (0%) | 2 (3%) | | |
| 16–20 years | 43 (74%) | 11 (19%) | 4 (7%) | 0 (0%) | 0 (0%) | | |
| ≥ 21 years | 217 (80%) | 39 (14%) | 3 (1%) | 2 (1%) | 11 (4%) | | |
| **Location of clinical practice** | | | | | | | |
| Rural area | 111 (72%) | 35 (23%) | 1 (1%) | 3 (2%) | 4 (3%) | 0.411 | 0.142 |
| Urban area | 330 (80%) | 62 (15%) | 6 (1%) | 2 (<1%) | 15 (4%) | | |
| **Everyone should be receiving the recommended vaccinations (excluding those with prohibiting medical conditions)** | | | | | | | |
| **Practice Setting** | | | | | | | |
| Private Practice | 175 (47%) | 127 (34%) | 30 (8%) | 25 (7%) | 16 (4%) | 0.094 | 0.058 |
| Academic Institution | 49 (59%) | 21 (25%) | 4 (5%) | 3 (4%) | 6 (7%) | | |
| FQHC and similar * | 41 (56%) | 24 (33%) | 1 (1%) | 5 (7%) | 2 (3%) | | |
| DSO ** | 10 (56%) | 1 (6%) | 3 (17%) | 1 (6%) | 3 (17%) | | |
| Other *** | 16 (73%) | 5 (23%) | 0 (0%) | 0 (0%) | 1 (5%) | | |
| **Years in Practice** | | | | | | | |
| 0–5 years | 56 (57%) | 25 (25%) | 6 (6%) | 6 (6%) | 6 (6%) | 0.761 | 0.248 |
| 6–10 years | 19 (27%) | 36 (51%) | 9 (13%) | 3 (4%) | 3 (4%) | | |
| 11–15 years | 38 (54%) | 29 (41%) | 1 (1%) | 2 (3%) | 0 (0%) | | |
| 16–20 years | 33 (57%) | 7 (12%) | 5 (9%) | 10 (17%) | 3 (5%) | | |
| ≥ 21 years | 145 (53%) | 81 (30%) | 17 (6%) | 13 (5%) | 16 (6%) | | |
| **Location of clinical practice** | | | | | | | |
| Rural area | 66 (43%) | 55 (36%) | 11 (7%) | 15 (10%) | 7 (5%) | 0.047 | 0.040 |
| Urban area | 225 (54%) | 123 (30%) | 27 (7%) | 19 (5%) | 21 (5%) | | |
| **Given the COVID-19 pandemic, if the Indiana State Board of Dentistry authorizes dentists to provide vaccinations would you consider offering vaccination in your practice?** | | | | | | | |
| **Practice Setting** | | | | | | | |
| Private Practice | 103 (28%) | 117 (31%) | 70 (19%) | 36 (10%) | 47 (13%) | 0.039 | 0.014 |
| Academic Institution | 28 (34%) | 28 (34%) | 13 (16%) | 4 (5%) | 10 (12%) | | |
| FQHC and similar * | 31 (42%) | 24 (33%) | 8 (11%) | 1 (1%) | 9 (12%) | | |
| DSO ** | 4 (22%) | 4 (22%) | 3 (17%) | 2 (11%) | 5 (28%) | | |
| Other *** | 10 (45%) | 1 (5%) | 7 (32%) | 3 (14%) | 1 (5%) | | |
| **Years in Practice** | | | | | | | |
| 0–5 years | 31 (31%) | 32 (32%) | 16 (16%) | 3 (3%) | 17 (17%) | 0.886 | 0.357 |
| 6–10 years | 14 (20%) | 30 (43%) | 12 (17%) | 12 (17%) | 2 (3%) | | |
| 11–15 years | 12 (17%) | 33 (47%) | 14 (20%) | 7 (10%) | 4 (6%) | | |
| 16–20 years | 26 (45%) | 12 (21%) | 9 (16%) | 2 (3%) | 9 (16%) | | |
| ≥ 21 years | 93 (34%) | 67 (25%) | 50 (18%) | 22 (8%) | 40 (15%) | | |
| **Location of clinical practice** | | | | | | | |
| Rural area | 52 (34%) | 38 (25%) | 28 (18%) | 22 (14%) | 14 (9%) | 0.940 | 0.732 |
| Urban area | 124 (30%) | 136 (33%) | 73 (18%) | 24 (6%) | 58 (14%) | | |
| **Dental providers are competent enough to be able to administer vaccines and need no further education/training** | | | | | | | |
| **Practice Setting** | | | | | | | |
| Private Practice | 95 (25%) | 102 (27%) | 70 (19%) | 79 (21%) | 27 (7%) | 0.330 | 0.080 |

*(Continued)*

**Table 4.** (Continued)

| | | | | | | | |
|---|---|---|---|---|---|---|---|
| Academic Institution | 28 (34%) | 28 (34%) | 7 (8%) | 15 (18%) | 5 (6%) | | |
| FQHC and similar * | 13 (18%) | 25 (34%) | 13 (18%) | 19 (26%) | 3 (4%) | | |
| DSO ** | 10 (56%) | 0 (0%) | 1 (6%) | 6 (33%) | 1 (6%) | | |
| Other *** | 9 (41%) | 3 (14%) | 3 (14%) | 6 (27%) | 1 (5%) | | |
| **Years in Practice** | | | | | | | |
| 0–5 years | 30 (30%) | 24 (24%) | 19 (19%) | 19 (19%) | 7 (7%) | 0.016 | 0.004 |
| 6–10 years | 29 (41%) | 13 (19%) | 7 (10%) | 21 (30%) | 0 (0%) | | |
| 11–15 years | 19 (27%) | 29 (41%) | 11 (16%) | 11 (16%) | 0 (0%) | | |
| 16–20 years | 14 (24%) | 22 (38%) | 8 (14%) | 8 (14%) | 6 (10%) | | |
| ≥ 21 years | 63 (23%) | 70 (26%) | 49 (18%) | 66 (24%) | 24 (9%) | | |
| **Location of clinical practice** | | | | | | | |
| Rural area | 49 (32%) | 44 (29%) | 17 (11%) | 35 (23%) | 9 (6%) | 0.231 | 0.082 |
| Urban area | 106 (26%) | 114 (27%) | 77 (19%) | 90 (22%) | 28 (7%) | | |
| **HPV related Oro-pharyngeal cancers can be prevented by use of vaccines** | | | | | | | |
| **Practice Setting** | | | | | | | |
| Private Practice | 143 (38%) | 146 (39%) | 71 (19%) | 3 (1%) | 10 (3%) | 0.003 | < .001 |
| Academic Institution | 29 (35%) | 37 (45%) | 15 (18%) | 2 (2%) | 0 (0%) | | |
| FQHC and similar * | 38 (52%) | 27 (37%) | 6 (8%) | 1 (1%) | 1 (1%) | | |
| DSO ** | 13 (72%) | 2 (11%) | 3 (17%) | 0 (0%) | 0 (0%) | | |
| Other *** | 17 (77%) | 2 (9%) | 3 (14%) | 0 (0%) | 0 (0%) | | |
| **Years in Practice** | | | | | | | |
| 0–5 years | 56 (57%) | 26 (26%) | 14 (14%) | 0 (0%) | 3 (3%) | 0.072 | 0.126 |
| 6–10 years | 25 (36%) | 34 (49%) | 10 (14%) | 1 (1%) | 0 (0%) | | |
| 11–15 years | 24 (34%) | 30 (43%) | 16 (23%) | 0 (0%) | 0 (0%) | | |
| 16–20 years | 28 (48%) | 12 (21%) | 18 (31%) | 0 (0%) | 0 (0%) | | |
| ≥ 21 years | 107 (39%) | 112 (41%) | 40 (15%) | 5 (2%) | 8 (3%) | | |
| **Location of clinical practice** | | | | | | | |
| Rural area | 64 (42%) | 55 (36%) | 32 (21%) | 3 (2%) | 0 (0%) | 0.978 | 0.972 |
| Urban area | 176 (42%) | 159 (38%) | 66 (16%) | 3 (1%) | 11 (3%) | | |
| **It would be easier for patients to complete their HPV vaccine schedule if they were to receive it from their dentists** | | | | | | | |
| **Practice Setting** | | | | | | | |
| Private Practice | 68 (18%) | 148 (40%) | 112 (30%) | 21 (6%) | 24 (6%) | 0.478 | 0.380 |
| Academic Institution | 19 (23%) | 27 (33%) | 29 (35%) | 2 (2%) | 6 (7%) | | |
| FQHC and similar * | 26 (36%) | 11 (15%) | 22 (30%) | 10 (14%) | 4 (5%) | | |
| DSO ** | 9 (50%) | 1 (6%) | 8 (44%) | 0 (0%) | 0 (0%) | | |
| Other *** | 5 (23%) | 8 (36%) | 6 (27%) | 3 (14%) | 0 (0%) | | |
| **Years in Practice** | | | | | | | |
| 0–5 years | 33 (33%) | 27 (27%) | 26 (26%) | 10 (10%) | 3 (3%) | 0.001 | < .001 |
| 6–10 years | 16 (23%) | 36 (51%) | 9 (13%) | 6 (9%) | 3 (4%) | | |
| 11–15 years | 14 (20%) | 29 (41%) | 23 (33%) | 4 (6%) | 0 (0%) | | |
| 16–20 years | 17 (29%) | 16 (28%) | 18 (31%) | 2 (3%) | 5 (9%) | | |
| ≥ 21 years | 47 (17%) | 87 (32%) | 101 (37%) | 14 (5%) | 23 (8%) | | |
| **Location of clinical practice** | | | | | | | |
| Rural area | 30 (19%) | 65 (42%) | 46 (30%) | 8 (5%) | 5 (3%) | 0.235 | 0.213 |
| Urban area | 97 (23%) | 130 (31%) | 131 (32%) | 28 (7%) | 29 (7%) | | |

FQHC* = Federally qualified health center

DSO** = Dental Service Organization

Others*** = Hospital Based Clinic, Local Health Department, Mobile Dentistry Practice, Other (not specified)

agreement than dentists at any other practice type, and dentists in private practice had a lower level of agreement than dentists at an academic institution. In contrast, the level of agreement with "I am comfortable administering vaccines in adults" was not significantly associated with years of practice, dentists working at DSOs had a higher level of agreement than dentists at Federally Qualified Health Centers (FQHCs), while dentists at an academic institution had a higher level of agreement than dentists in private practice or FQHCs.

The level of agreement with "Dentists should be allowed to administer HPV, Influenza, Hep A and COVID 19" for both children and adults decreased with increased years of practice. Dentists working at DSOs had a higher level of agreement than dentists in private practice, academic institutions, or FQHCs.

Dentists practicing in a rural area had a lower level of agreement than dentists in urban area for "Everyone should be receiving the recommended vaccinations (excluding those with prohibiting medical conditions)". Dentists working at DSOs had lower level agreement with "consider offering vaccination in your practice" than dentists at academic institutions and FQHCs, and dentists in private practice had lower level of agreement than dentists at FQHCs. The level of agreement with "Dental providers are competent enough to be able to administer vaccines and need no further education/training" and "It would be easier for patients to complete their HPV vaccine schedule if they were to receive it from their dentists" with decreased years of practice. Dentists working in private practice had lower level of agreement with "HPV related Oro-pharyngeal cancers can be prevented by use of vaccines" than dentists practicing at FQHCs, DSOs, and other practice types, and dentists working at academic institutions had lower level of agreement than dentists practicing at DSOs and other practice types.

### Barriers (not referenced in tables)

Reported challenges in being able to administer vaccines included the following categories (among all respondents): storage of vaccines/supply chain (77%), reimbursement (71%), insufficient training/knowledge (56%), comfort levels (32%), time (28%) and role confusion (27%). Among private practitioners, 78% reported Storage of Vaccines/supply chain as the biggest challenge, again followed by reimbursement (73%). The participants who selected "other reasons" as challenges expressed concerns around their inability to handle anaphylactic reactions and side effects that may occur (though infrequently) during vaccine administration. For some, it was a concern of their scope of practice and the medicolegal concerns surrounding it.

### Discussion

Vaccination, one of the biggest public health achievements of the past century, has brought about a drastic reduction in morbidity and mortality rates, especially in the pediatric population [29–31]. The adolescent platform for vaccination is still being explored to develop and evaluate interventions to increase uptake of adolescent vaccines [32]. However, little to no data are available on dentists and vaccination administration. Oral health providers have been wary of their scope of practice, especially in terms of being able to administer vaccines, for a variety of reasons. Lack of adequate training, storage and handling, associated cost inefficiency and current billing practices are some of the reasons why dentists do not find administering vaccines very feasible [33]. However, it is noteworthy that in the past, oral health professionals have been successfully involved in several preventive care campaigns (e.g., tobacco cessation, blood pressure and glucose monitoring, and oral cancer screenings). Their role in primary health care preventive strategies, including vaccinations, should be explored further [34].

The study results strongly suggest the willingness of dentists in the state of Indiana to offer vaccination in their practice, if allowed by legislation. Per the results of only this study, 331

vaccinators could be added to the state's vaccination workforce with such legislation. These numbers are however, rough estimates based on the dentists' willingness to consider vaccinating their patients, rather than their overall readiness to vaccinate. More than half of our study respondents were private practitioners which can be attributed to the fact that most dentists (93.6%) in the state of Indiana, work in private practices [35]. The dentists practicing in FQHCs were more agreeable to offer vaccination in their practice as compared to those in private practice and working for DSOs. It is worth mentioning that FQHCs have both medical and dental personnel working within the same facility so there may lesser value in having dentists administering vaccines. However, the reason behind this still being a great concept is because having more providers capable of administering vaccines will only improve vaccination efforts, especially if they practiced in an alternate setting. FQHCs could be an ideal example for dentists to take a lead on this as a lot of concerns related to the handling and storage of vaccines, may be better acnowledged (and less expensive) in this setting as compared to private dental offices. Additionally, FQHCs typically have an integrated medical-dental setting, which would allow for addressing side effects or anaphylactic reactions to the vaccine by nearby medical professionals.

It is also worth mentioning that the dental providers were more comfortable with being able to vaccinate adults as compared to children. The reasons behind that may be: reimbursement/insurance concerns, comfort levels of the providers, and the fact that child vaccination is often coupled with a "well-child visit," which may be more preferred by the parents or caregivers. The willingness of dentists to vaccinate adults over children could be a concerning issue that needs to be addressed, especially for vaccines like HPV that have shown diminishing effectiveness in older ages [18]. Younger dentists or those new in practice were more comfortable administering vaccines in children as compared to those who had been in practice for longer. Most respondents did report in the open-ended comments section their preference for required training or continuing education courses on vaccine administration for dental providers.

The American Academy of Pediatric Dentistry and American Dental Association recognize the role of dental providers and support measures that prevent oropharyngeal cancers, including the prevention of HPV infection [36, 37]. With added workforce capacity by including dentists as vaccinators and educators, it may be easier to reach the Healthy People 2020's proposed goal of HPV vaccine uptake of 80% [19]. This might prove to be very effective for HPV-related oropharyngeal cancer prevention initiatives (along with other cancers), which corresponds well with the recent FDA approval of the HPV vaccine (Gardasil 9) for oropharyngeal cancer prevention [38–40].

Subject literacy, stigma around the topic, reimbursement, and parents' hesitation are a few reasons why dentists are reluctant to even discuss the HPV vaccination topic with their patients/caregivers [3, 41, 42]. However, with adequate training and recognition of their role in HPV-related oropharyngeal cancer prevention, overcoming such challenges may be easier.

Our study results were comparable to a previous study on "less typical" healthcare providers (pharmacists) as vaccinators; majority of the patients receiving vaccines were adults and the major challenges reported were reimbursement, lack of information about vaccines and adverse reactions associated [43]. Other studies have also reported positive attitudes towards pharmacists administering vaccines [28, 44]. Pharmacists are now a major part of the COVID-19 vaccination efforts and the above such studies well predicted the current COVID-19 vaccination practices which include other providers (eg, pharmacists).

### Strengths and limitations

This study included participants from various practice models, including private practice, federally qualified health centers, dental service organizations, and academia. The results provide a good understanding of how the providers felt about administering vaccinations with respect to the type of their practice. Also, included in our respondents were providers who had been in practice for more than twenty years and those who were relatively new providers, offering insight into the differences in opinions by years of experience. Such information may be useful in devising policy changes to alter the scope of practice for dentists in Indiana and other states.

This study was conducted during the COVID-19 pandemic. As the literature suggests, dentists have been part of vaccine administration during crises in the past, so the survey responses may suggest a current sense of obligation by participants to assist in vaccine rollout to help frontline workers and public health initiatives. This may not persist under normal times in their general scope of practice. The survey questions also did not separate out the different vaccines, so it is possible that providers may be comfortable offering some vaccines more than others. The cross sectional study design which had self reported information may also have social desirability bias [45] which could limit the validity of the study results. The survey responses considered in the analyses were 569, which is approximately 15% of the total number of dentists (4020) with an active license in Indiana per the reports from the Bowen Center for Health Workforce Research and Policy [35]. This also implies, 85% of the dentists in the state, did not contribute data to the study thus creating significant response bias. It's important to note that the way the survey was worded- "consider offering vaccination" may have a different meaning from "willingness" or "readiness", reflecting a potential weakness in the survey design.

Although this study was based on a survey of dentists' willingness to administer vaccines in the state of Indiana, it can be relevant nationwide. With current United States' federal regulations allowing dental providers to administer COVID-19 vaccines and employing the past example of the H1N1 pandemic, the time is definitely ripe to advocate for inclusion of vaccine administration in the scope of practice of dental providers nationally [23, 46]. Recently in Indiana, the Indiana Senate unanimously approved HB 1079, allowing dentists with proper training to administer vaccines [47].

Further research exploring non- traditional settings to overcome the infrastructure challenges of supply chain, storage and cost inefficiency may be helpful in improving participation of dental providers, especially in private practice settings. Policies addressing reimbursement and liability issues also need to be addressed to convince more dentists of their role in vaccine administration.

## Supporting information

**S1 File.**
(DOCX)

## Author Contributions

**Conceptualization:** Anubhuti Shukla, Kelly Welch, Alessandro Villa.

**Data curation:** Anubhuti Shukla.

**Formal analysis:** Anubhuti Shukla.

**Funding acquisition:** Anubhuti Shukla, Kelly Welch.

**Investigation:** Anubhuti Shukla, Kelly Welch, Alessandro Villa.

**Methodology:** Anubhuti Shukla, Alessandro Villa.

**Project administration:** Anubhuti Shukla.

**Resources:** Anubhuti Shukla, Kelly Welch, Alessandro Villa.

**Software:** Anubhuti Shukla.

**Supervision:** Anubhuti Shukla, Alessandro Villa.

**Validation:** Anubhuti Shukla.

**Visualization:** Anubhuti Shukla.

**Writing – original draft:** Anubhuti Shukla.

**Writing – review & editing:** Anubhuti Shukla, Kelly Welch, Alessandro Villa.

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
