## [Decision Letter · Decision Letter 0]

26 Dec 2021

PONE-D-21-36322Assessment of the Willingness of Dentists in the State of Indiana to administer vaccines.PLOS ONE

Dear Dr. SHUKLA,

Thank you for submitting your manuscript to PLOS ONE. After careful consideration, we feel that it has merit but does not fully meet PLOS ONE’s publication criteria as it currently stands. Therefore, we invite you to submit a revised version of the manuscript that addresses the points raised during the review process.

We look forward to receiving your revised manuscript.

Kind regards,

David M. Ojcius

Academic Editor

PLOS ONE

Journal Requirements:

Reviewers' comments:

Reviewer's Responses to Questions

**Comments to the Author**

1. Is the manuscript technically sound, and do the data support the conclusions?

Reviewer #1: No

Reviewer #2: Yes

2. Has the statistical analysis been performed appropriately and rigorously? 

Reviewer #1: I Don't Know

Reviewer #2: I Don't Know

3. Have the authors made all data underlying the findings in their manuscript fully available?

Reviewer #1: No

Reviewer #2: Yes

4. Is the manuscript presented in an intelligible fashion and written in standard English?

Reviewer #1: No

Reviewer #2: Yes

5. Review Comments to the Author

Reviewer #1: I have reviewed the manuscript, "Assessment of the Willingness of Dentists in the State of Indiana to administer vaccines" submitted to PlosOne for its contents. Authors have investigated the readiness of dentists in the state of Indiana in being able to administer vaccines.

This manuscript does not present enough data and novelty for a full length original article.

Reviewer #2: I enjoyed very much reading this this well written, well-structured manuscript that provides a valuable contribution to the literature about the role of dentists in administering vaccinations as members of the wider healthcare team.

I am very happy with the paper in this current stage, and I hope that the authors will find my comments below useful to strengthen the paper even more.

Abstract

“Dentists should be allowed to administer HPV, Influenza, Hep A and COVID 19” I think the word “vaccinations” might be missing from that sentence.

Background

“the Public Readiness and Emergency Preparedness (PREP) Act” is this specific for the state of Indiana or is it at US level?

The authors are presenting a comprehensive overview of the vaccinations that are being provided by dentists in various US states. As this is an international journal, it might strengthen the argument to mention a few examples of other countries where dentists have been allowed to provide vaccinations e.g., the UK.

“Oropharyngeal cancers have been on the rise in the past three decades due to persistent high-risk-type human papillomavirus (HPV) infections”. Whilst I agree with the first part of the statement, that oropharyngeal cancers have been on the rise in the past three decades, I believe that it might be difficult to attribute this to a singular cause e.g., increased prevalence of HPV infections. It might be worth considering also mentioning the increasing role of the wider risk factors e.g., alcohol and smoking.

“Pre-COVID 19 pandemic averages of HPV vaccine completion rates in the US were low (54.2%)”. Does this include both boys and girls or girls only?

“Expanding dental providers scope of practice to include vaccination could provide similar results as with pharmacists, however such changes nationwide would call for legislative enabling.” Would there be anything else needed? E.g., professional indemnity/insurance, training?

Results

“More than half of the respondents (68%) reported having a policy for oral cancer screening in their office.” I couldn’t find this data in Table 1.

I noticed that the authors have presented in several places p values to attribute statistical significance. I am wondering if this really necessary and if it adds value to the paper. A significant number of journals have been moving more towards “confidence intervals” instead of p values to inform readers about the precision of the results as a more robust measure than p values. Also, I noticed the number of participants in some of the subgroup analysis is quite small therefore some of the statistical significance might be due to chance. I am not a statistician, and I am happy to be challenged here but perhaps a sample size calculation might have been useful to avoid the risk of multiple testing bias. I think perhaps percentages and/or confidence intervals might be an alternative to consider by the authors.

The authors talk about “the level of agreement”. I am not sure if I understand this correctly. Is this level of agreement based on percentage of responses agreeing with a statement or based on the p values?

Table 3 and table 4 presents two columns: single variable and multivariable. The authors might wish to consider clarifying what are the variables.

Barriers (not referenced in tables).

Does the first sentence refer to all respondents in general or only dentists working in the public sector? The second sentence starts with “private practitioners”. Might be worth clarifying which group does the first sentence refer to.

Discussion

The authors argue about the opportunities of using dentists for administering vaccinations. Whilst this is important and noteworthy, it might be worth considering that dental hygienists/therapists and trained dental nurses might also have a role, which might be even more cost effective. Not sure about the situation in the US but in some countries, flu vaccinations in GP surgeries are not always administered by GPs but by trained nurses. This frees up GP time to deal with day-to-day business that requires a qualified physician. Might be worth considering the implications of using the wider dental workforce, not just dentists, to administer vaccinations as long as they are properly trained, competent, indemnified and remunerated.

“More than half of our study respondents were private practitioners which is comparable to the practice distribution of dental providers in the state of Indiana, where most dentists (93.6%) work in private practices.” Does this sentence imply that the dentists working in public sector were over represented in the study?

FQHCs and DSOs appear in multiple places. Might be worth repeating what these acronyms mean for a the non-US based readers.

“Further research exploring non- traditional settings to overcome the infrastructure challenges of supply chain, storage and cost inefficiency may be helpful in improving participation of dental providers, especially in private practice settings.” Does cost inefficiency mean cost effectiveness in this context?

Once again I congratulate the authors for this important research and I am looking forward to reading their published article.

6. PLOS authors have the option to publish the peer review history of their article (what does this mean?). If published, this will include your full peer review and any attached files.

Reviewer #1: No

Reviewer #2: No

---

## [Author Response · Author response to Decision Letter 0]

3 Jan 2022

1/3/2022

Title: Assessment of the Readiness of Dentists in the State of Indiana to administer vaccines.   

Dear Editors,

We would like to thank the reviewers for their valuable time and effort in reviewing our manuscript. We have addressed all the comments and made edits in the blinded manuscript suing track changes as recommended. 

Please find our responses to each comment and suggestions below, for additional clarification. 

I remain available for any questions.

Thanks again.

Dr. Anubhuti Shukla

REVIEWER’S COMMENTS

Abstract

“Dentists should be allowed to administer HPV, Influenza, Hep A and COVID 19” I think the word “vaccinations” might be missing from that sentence.

Response: Thanks for catching that. The term “vaccines” has been added as suggested.

Background

“the Public Readiness and Emergency Preparedness (PREP) Act” is this specific for the state of Indiana or is it at US level?

Response: This is a national level emergency authorization

https://www.phe.gov/Preparedness/legal/prepact/Pages/default.aspx

The authors are presenting a comprehensive overview of the vaccinations that are being provided by dentists in various US states. As this is an international journal, it might strengthen the argument to mention a few examples of other countries where dentists have been allowed to provide vaccinations e.g., the UK.

Response: Thanks for your comment. We included a new paragraph in the introduction and reported on the global data on COVID-19 vaccinations and the role of the dental community. (Paragraph 1)

“Oropharyngeal cancers have been on the rise in the past three decades due to persistent high-risk-type human papillomavirus (HPV) infections”. Whilst I agree with the first part of the statement, that oropharyngeal cancers have been on the rise in the past three decades, I believe that it might be difficult to attribute this to a singular cause e.g., increased prevalence of HPV infections. It might be worth considering also mentioning the increasing role of the wider risk factors e.g., alcohol and smoking.

Response: Out study was conducted in the US where the increase in the incidence of oropharyngeal cancers is mainly attributed to HPV high risk infections. We modified the sentence to make this clearer.

“Pre-COVID 19 pandemic averages of HPV vaccine completion rates in the US were low (54.2%)”. Does this include both boys and girls or girls only? 

Response: yes, it includes both boys and girls

“Expanding dental providers scope of practice to include vaccination could provide similar results as with pharmacists, however such changes nationwide would call for legislative enabling.” Would there be anything else needed? E.g., professional indemnity/insurance, training? 

Response: yes, we agree that educational training and professional indemnity may be needed. We modified the sentence as suggested by the reviewer. 

Results

“More than half of the respondents (68%) reported having a policy for oral cancer screening in their office.” I couldn’t find this data in Table 1.

Response: Since this was a question not directly related to vaccinations, it wasn’t included in the table (data not shown).

I noticed that the authors have presented in several places p values to attribute statistical significance. I am wondering if this really necessary and if it adds value to the paper. A significant number of journals have been moving more towards “confidence intervals” instead of p values to inform readers about the precision of the results as a more robust measure than p values. Also, I noticed the number of participants in some of the subgroup analysis is quite small therefore some of the statistical significance might be due to chance. I am not a statistician, and I am happy to be challenged here but perhaps a sample size calculation might have been useful to avoid the risk of multiple testing bias. I think perhaps percentages and/or confidence intervals might be an alternative to consider by the authors.

Response: Although confidence intervals can be a bit more robust than p-values for interpreting significance of results, adding confidence intervals to results from surveys such as this are difficult to present in a way that’s easy for the reader to follow – as an example, for each of the 50 percentages shown in Table 2 there would be an additional 2 percentages shown for each one if the confidence intervals are added – so it would have the 50 N’s, 50 calculated percentages, and 100 additional percentages representing the confidence intervals – for the first piece of Table 2, instead of ‘441 (78%)’, the table would show 441 (78%, 74%-81%). We feel that the tradeoff using the p-values to show the results concisely outweigh the additional information added by the confidence intervals.

Sample size calculations performed prior to conducting a survey are rarely informative. The actual response rates are unknown and expected response rates vary widely. Further, the calculations depend not only on the unknown expected response rate but also on the distributions of the responses to the individual questions (also unknown prior to data collection).

The authors talk about “the level of agreement”. I am not sure if I understand this correctly. Is this level of agreement based on percentage of responses agreeing with a statement or based on the p values?

Response: Thanks for the question. This was based on the number (%) of responses

Table 3 and table 4 presents two columns: single variable and multivariable. The authors might wish to consider clarifying what are the variables.

Response: Single-variable refers to statistical tests using only the specific characteristic. Multivariable refers to statistical tests that include all three characteristics (practice setting, years in practice, location of clinical practice) simultaneously in the same statistical model.

Barriers (not referenced in tables).

Does the first sentence refer to all respondents in general or only dentists working in the public sector? The second sentence starts with “private practitioners”. Might be worth clarifying which group does the first sentence refer to.

Response: We added the clarification as suggested.

Discussion

The authors argue about the opportunities of using dentists for administering vaccinations. Whilst this is important and noteworthy, it might be worth considering that dental hygienists/therapists and trained dental nurses might also have a role, which might be even more cost effective. Not sure about the situation in the US but in some countries, flu vaccinations in GP surgeries are not always administered by GPs but by trained nurses. This frees up GP time to deal with day-to-day business that requires a qualified physician. Might be worth considering the implications of using the wider dental workforce, not just dentists, to administer vaccinations as long as they are properly trained, competent, indemnified and remunerated.

Response: We completely agree that dental hygienists may play a role in vaccine administration. However, the role of mid-level providers in dentistry is very restrictive in the US. Therefore, while it makes complete sense, including dental hygienists and therapists will take much longer and a different advocacy effort.

“More than half of our study respondents were private practitioners which is comparable to the practice distribution of dental providers in the state of Indiana, where most dentists (93.6%) work in private practices.” Does this sentence imply that the dentists working in public sector were over represented in the study?

Response: Since the sampling method was random, we did not oversample any groups, the sentence simply states that the distribution of respondents is pretty similar to the dental practitioners’ distribution in Indiana.

FQHCs and DSOs appear in multiple places. Might be worth repeating what these acronyms mean for a the non-US based readers.

Response: they are both referenced on page 12, last paragraph.

“Further research exploring non- traditional settings to overcome the infrastructure challenges of supply chain, storage and cost inefficiency may be helpful in improving participation of dental providers, especially in private practice settings.” Does cost inefficiency mean cost effectiveness in this context?

Response: cost inefficiency refers to the vaccine storage investment needed vs the reimbursement for administering the vaccine to patients in the dental office.

Thank you again for your helpful comments and suggestions. Please let us know if you have any additional questions.

---

## [Decision Letter · Decision Letter 1]

7 Feb 2022

PONE-D-21-36322R1Assessment of the Willingness of Dentists in the State of Indiana to administer vaccines.PLOS ONE

Dear Dr. SHUKLA,

Thank you for submitting your manuscript to PLOS ONE. After careful consideration, we feel that it has merit but does not fully meet PLOS ONE’s publication criteria as it currently stands. Therefore, we invite you to submit a revised version of the manuscript that addresses the points raised during the review process.

We look forward to receiving your revised manuscript.

Kind regards,

David M. Ojcius

Academic Editor

PLOS ONE

Reviewers' comments:

Reviewer's Responses to Questions

**Comments to the Author**

1. If the authors have adequately addressed your comments raised in a previous round of review and you feel that this manuscript is now acceptable for publication, you may indicate that here to bypass the “Comments to the Author” section, enter your conflict of interest statement in the “Confidential to Editor” section, and submit your "Accept" recommendation.

Reviewer #1: (No Response)

Reviewer #3: All comments have been addressed

2. Is the manuscript technically sound, and do the data support the conclusions?

Reviewer #1: No

Reviewer #3: Yes

3. Has the statistical analysis been performed appropriately and rigorously? 

Reviewer #1: No

Reviewer #3: Yes

4. Have the authors made all data underlying the findings in their manuscript fully available?

Reviewer #1: No

Reviewer #3: Yes

5. Is the manuscript presented in an intelligible fashion and written in standard English?

Reviewer #1: Yes

Reviewer #3: Yes

6. Review Comments to the Author

Reviewer #1: (No Response)

Reviewer #3: This study investigates a relevant question, and answers it with details that can guide policymakers for vaccination efforts. I have some minor recommendations:

1. It would be helpful to give a rough estimate of how many vaccinators may be added to the state's vaccination workforce if dentists are allowed to vaccinate, based on the percentage of dentists in the study that said they would consider vaccinating if allowed. That calculation can be given in the form of the number of dentists that may join the vaccination workforce, as well as what percent that may increase the vaccination workforce. These values would have to be tempered with an admission that these are purely rough estimates based on willingness to consider vaccinating, rather than readiness to vaccinate.

2. In the Introduction, 2nd paragraph, 2nd sentence, the statement "More than half of our study respondents were private practitioners which is comparable to the practice distribution of dental providers in the state of Indiana, where most dentists (93.6%) work in private practices" is hard to rectify. >50% and 93.6% seem to far apart to be considered comparable.

3. In a FQHC, which will likely already have medical personnel administering vaccines in the same building as a dental clinic, there may be limited value to having dentists administer vaccines. I recommend at least mentioning this, as well as any reason why it may still be beneficial to have dentists in FQHCs administering vaccines.

4. The authors reference a previous study that looked at pharmacists' willingness to vaccinate. Now, pharmacists are a major part of the COVID vaccination effort. It would be valuable to mention how well that study predicted the current stage of the COVID vaccination drive and pharmacists' involvement.

5. Please explain what is meant by the term "social desirability" in page 18, top paragraph.

6. Please check if more states now permit dentists to vaccinate, since the original submission of this manuscript.

7. PLOS authors have the option to publish the peer review history of their article (what does this mean?). If published, this will include your full peer review and any attached files.

Reviewer #1: No

Reviewer #3: No

---

## [Author Response · Author response to Decision Letter 1]

14 Feb 2022

All requested changes have been made to the manuscript, one with tracked changes shows all the edits. the rebuttal document also expands on all the requested changes. thank you to the reviewers for their time.

---

## [Decision Letter · Decision Letter 2]

4 Apr 2022

Assessment of the willingness of dentists in the state of Indiana to administer vaccines.

PONE-D-21-36322R2

Dear Dr. Shukla,

We’re pleased to inform you that your manuscript has been judged scientifically suitable for publication and will be formally accepted for publication once it meets all outstanding technical requirements.

Kind regards,

David M. Ojcius

Academic Editor

PLOS ONE

Reviewers' comments:

Reviewer's Responses to Questions

**Comments to the Author**

1. If the authors have adequately addressed your comments raised in a previous round of review and you feel that this manuscript is now acceptable for publication, you may indicate that here to bypass the “Comments to the Author” section, enter your conflict of interest statement in the “Confidential to Editor” section, and submit your "Accept" recommendation.

Reviewer #3: All comments have been addressed

2. Is the manuscript technically sound, and do the data support the conclusions?

Reviewer #3: Yes

3. Has the statistical analysis been performed appropriately and rigorously? 

Reviewer #3: Yes

4. Have the authors made all data underlying the findings in their manuscript fully available?

Reviewer #3: Yes

5. Is the manuscript presented in an intelligible fashion and written in standard English?

Reviewer #3: Yes

6. Review Comments to the Author

Reviewer #3: My recommendations have been adequately addressed. The manuscript has adequate study design, and is clearly explained.

7. PLOS authors have the option to publish the peer review history of their article (what does this mean?). If published, this will include your full peer review and any attached files.

Reviewer #3: **Yes: **Andrew Young

---

## [Editor Report · Acceptance letter]

8 Apr 2022

PONE-D-21-36322R2 

Assessment of the willingness of dentists in the state of Indiana to administer vaccines. 

Dear Dr. Shukla:

I'm pleased to inform you that your manuscript has been deemed suitable for publication in PLOS ONE. Congratulations! Your manuscript is now with our production department. 

Kind regards, 

on behalf of

Dr. David M. Ojcius 

Academic Editor

PLOS ONE